# DNA methylation predicts infection risk in kidney transplant recipients

Fei-Man Hsu[1,2] ⓘ, Harry Pickering[3], Liudmilla Rubbi[1], Michael Thompson[1], Elaine F Reed[3], Matteo Pellegrini[1,2],† ⓘ, Joanna M Schaenman[4],† ⓘ

**Kidney transplantation (KTx) is the method of choice for treating kidney failure. Identifying biomarkers predictive of transplant (Tx) outcomes is critical to optimize KTx; however, the immunosuppressive therapies required after KTx must also be considered. We applied targeted bisulfite sequencing (TBS-seq) to PBMCs isolated from 90 patients, with samples collected pre- and post-Tx (day 90), to measure DNA methylation changes. Our findings indicate that the PBMC DNA methylome is significantly affected by induction immunosuppression with anti-thymocyte globulin (ATG). We discovered that the risk of infection can be predicted using DNA methylation profiles, but not gene expression profiles. Specifically, 515 CpG *loci* associated with 275 genes were significantly impacted by ATG induction, even after accounting for age, sex, and cell-type composition. Notably, ATG-associated hyper-methylation down-regulates genes critical for immune response. In conclusion, this clinical omics study reveals that the immunosuppressant ATG profoundly impacts the DNA methylome of KTx recipients and identifies biomarkers that could be used in pre-Tx screening of patients vulnerable to infection, thereby informing immunosuppression strategies post-Tx.**

## Introduction

*Kidney transplantation (KTx)* is the optimal treatment for kidney failure, which improves survival while maintaining the quality of life over dialysis (Hariharan et al, 2021). According to the Centers for Disease Control and Prevention, one in seven adults, about 35.5 million people, are estimated to have chronic kidney disease (CKD), and there were ~25,000 KTx in the United States in the year 2022. The aging population and the rising incidences of diseases such as diabetes and hypertension lead to increased incidence of CKD, as well as increased demand for KTx (Moeller et al, 2002). Long-term allograft and patient survival are limited by infections and chronic rejection (Martin-Gandul et al, 2015; Wu et al, 2021). Given the increased vulnerability to infection in older patients with demonstrated immune senescence, there is a currently unmet need in the field to measure immune function at the cellular level to determine the impact of immunosuppression and develop predictors of post-transplant infection.

*Anti-thymocyte globulin (ATG)* is the primary lymphodepleting induction regimen for preventing or treating acute rejection (AR) after solid organ transplantation (Swanson et al, 2002; Pearl et al, 2005; Lim et al, 2017). This purified immunoglobulin G from rabbits immunized with human T cells and their precursor thymocytes induces immune cell depletion, particularly T-cell senescence and exhaustion immediately through complement-dependent cell lysis and phagocytosis (Mohty, 2007; Bamoulid et al, 2017). During ATG production, the cell types present in normal human thymus, for example, thymocytes, were used to inoculate rabbits in the production of thymoglobulin. T lymphocyte (72%) dominates the cell-type composition, with B cells (6%), APC, and stromal cells (22%) contributing to the rest (Zand et al, 2005). This antibody-dependent lymphodepletion mechanism targets cells expressing antigens that are used to prepare ATG in a dosage-dependent manner (Mohty, 2007), and it has been reported to result in inversion of the CD4[+]/ CD8[+] ratio after ATG treatment with a decline in the frequency of naïve CD4[+] and CD8[+] T cells (Mourad et al, 2012). T-cell reconstitution by homeostatic lymphocyte proliferation occurs by around 6 mo, depending on the patients' age and baseline T-cell numbers (Gurkan et al, 2010). Impaired CD4[+] T-cell reconstitution after ATG induction is a major cause of morbidity and mortality and leads to opportunistic infections and atherosclerosis in KTx recipients (Ducloux et al, 2010; Pham et al, 2020). Previous studies suggested that ATG has a greater impact on older individuals (>= 65 yr old) because of T-cell senescence that impedes lymphocyte proliferation (Krenzien et al, 2015). While remaining a powerful immunosuppressant for induction and treatment of rejection, ATG's long-term consequences and the optimal dosing scheme need to be further explored to optimize the benefits over risks. This is

---

[1]Department of Molecular, Cell and Developmental Biology, University of California Los Angeles, Los Angeles, CA, USA  [2]Institute for Quantitative and Computational Biosciences – The Collaboratory, University of California Los Angeles, Los Angeles, CA, USA  [3]Department of Pathology and Laboratory Medicine, David Geffen School of Medicine, University of California Los Angeles, Los Angeles, CA, USA  [4]Department of Medicine, David Geffen School of Medicine, University of California Los Angeles, Los Angeles, CA, USA

Correspondence: matteop@mcdb.ucla.edu; JSchaenman@mednet.ucla.edu
†Matteo Pellegrini and Joanna M Schaenman are joint senior authors

especially true for patients with older biological age as measured by DNA methylation, which was demonstrated in our previous research to be a better predictor of infection than chronologic ß (Schaenman et al, 2020). KTx recipients not receiving ATG typically receive induction of basiliximab (SIMULECT), an interleukin-2 receptor inhibitor without the lymphodepleting impact of ATG (Hill et al, 2017). Maintenance immunosuppression typically consists of tacrolimus (TAC), a calcineurin inhibitor, or belatacept (BELA), an inhibitor of the second signal, plus mycophenolate mofetil and prednisone (Lentine et al, 2021).

*DNA methylation* is an epigenetic modification associated with cumulative events such as environmental exposures, smoking, chronic diseases such as diabetes and its complications, neuro-degenerative diseases, cardiovascular diseases, and CKDs (Zhang et al, 2016; Wahl et al, 2017; Witasp et al, 2022; Smyth et al, 2023). It has been shown that the end-stage CKD patients have blood DNA methylation patterns associated with inflammation (Stenvinkel et al, 2007; Wing et al, 2014). Several lines of evidence suggest that DNA methylation is associated with post-renal transplant (post-Tx) complications such as ischemia–reperfusion injury (Zhao et al, 2017), graft fibrosis (Ko et al, 2013; Sagy et al, 2024), and alloimmune response (Braza et al, 2015; Hu et al, 2016; Cristoferi et al, 2022). However, the ability to predict the occurrence of these KTx complications remains poor, possibly because of limits of biopsies, prophylaxes, and high variability in human populations.

Here, we present a cohort with 90 subjects who received KTx at the Ronald Reagan Medical Center, at the University of California, Los Angeles. We used targeted bisulfite sequencing (TBS-seq) and RNA-seq to characterize the epigenome and transcriptome dynamics pre- and post-Tx. We trained machine learning models with DNA methylation to predict outcomes, and the results indicate that ATG induction and risk of infection could be predicted by DNA methylation. The predicted epi-infection score is significantly correlated to the time to infection within the 1-yr follow-up window in this cohort.

# Results

## DNA methylation is correlated with ATG induction

This study examined patients receiving KTx enrolled at the University of California, Los Angeles, between April 2015 and September 2021. The study design is shown in Fig 1A. Patients received immunosuppression medication immediately after KTx (day 0), including induction therapy (ATG or SIMULECT) during days 1–3 and long-term maintenance (TAC or BELA). All patients had prednisone doses tapered to 5 mg prednisone by mouth daily by 3 mo of KTx.

We collected patients' PBMCs on the day of KTx (pre-Tx) and ~90 d after the surgery (post-Tx) for TBS-seq and RNA-seq analyses. In total, 122 TBS-seq and 78 RNA-seq libraries were generated (Table 1). We further followed these patients in the post-Tx visits for 365 d and noted the occurrence of infection and acute rejection.

We first asked whether DNA methylation is associated with any demographic or clinical traits by performing principal component analysis (PCA). In addition to age and sex that are known to be associated with DNA methylation (Gatev et al, 2021), we found that

ATG induction is correlated with the PC1 axis, whereas Transplant (Pre/Post) and Infection Risk were not correlated with the top PCs (Fig 1B and C). This suggests that ATG induction impacts the recipient PBMC methylomes.

## ATG induction depletes CD4$^+$ and naïve T cells

Because ATG causes depletion of T-cell subtypes (Mohty, 2007), we sought to estimate T-cell composition within our samples from DNA methylation profiles. Immune cell-type methylomes were collected as references, and the coefficients of the constructed nonnegative least squares regression model were estimated as the fractions of specific cell types (see the Materials and Methods section). We found ATG induction reduced the fractions of non-naïve CD4$^+$ T-cell and the naïve T-cell populations, whereas the non-naïve CD8$^+$ T-cell fraction remained unchanged (Figs 2A and S1A and B).

The DNA methylation–estimated cell composition was validated by flow cytometry, which also showed that non-naïve CD4$^+$ and the naïve T cells are highly impacted by ATG compared to induction immunosuppression with SIMULECT (basiliximab), which acts as an interleukin-2 receptor antagonist blocking T-cell proliferation (Figs 2B and S1A and B) (Amlot et al, 1995). We note that our post-Tx samples were collected around 90 d after the KTx and the induction prophylaxes, suggesting that T-cell reconstitution is not completed by 90 d in this KTx cohort.

## Predicting ATG induction and infection risk with DNA methylation

Next, we sought to predict clinical outcomes with DNA methylation profiles and identify associated genomic *loci*. We built a penalized logistic regression model with CpG sites as covariates and the log odds of each trait as the dependent variable. When predicting whether the sample is collected pre-Tx or post-Tx, the area under the curve (AUC) was only 0.67 (Fig S2). ATG treatment is predictable from DNA methylation profiles with an AUC of 0.85 (Fig S2). The prediction of whether the individual is going to develop infection was not significant with an AUC of 0.52 (Fig S2). These results are consistent with the PCA results, showing that ATG is more strongly associated with DNA methylation principal components (PCs), compared with other variables such as transplant and infection.

Given the impact of ATG on cell composition, we used the estimated cell-type PCs together with other demographic and clinical outcomes to construct a multivariate multiple linear regression (MMLR) model (Fig 3A). This model considers the contribution per CpG *locus* of each trait. Using leave-one-out cross-validation, the refined MMLR model could predict ATG induction ($P < 0.001***$) and infection ($P < 0.001***$) (Fig 3B). ATG prediction has a 0.90 AUC, and infection has a 0.79 AUC (Fig 3C). These results suggest that ATG induction and vulnerability to infection are significantly associated with epigenetic profiles in addition to the standard covariates such as age, sex, CMV serostatus, and cell-type composition.

## Site-specific DNA methylation impact of ATG induction

We next asked which CpG *loci* are significantly associated with ATG induction or risk of infection. A multiple linear regression model was constructed per CpG to test for association, while controlling

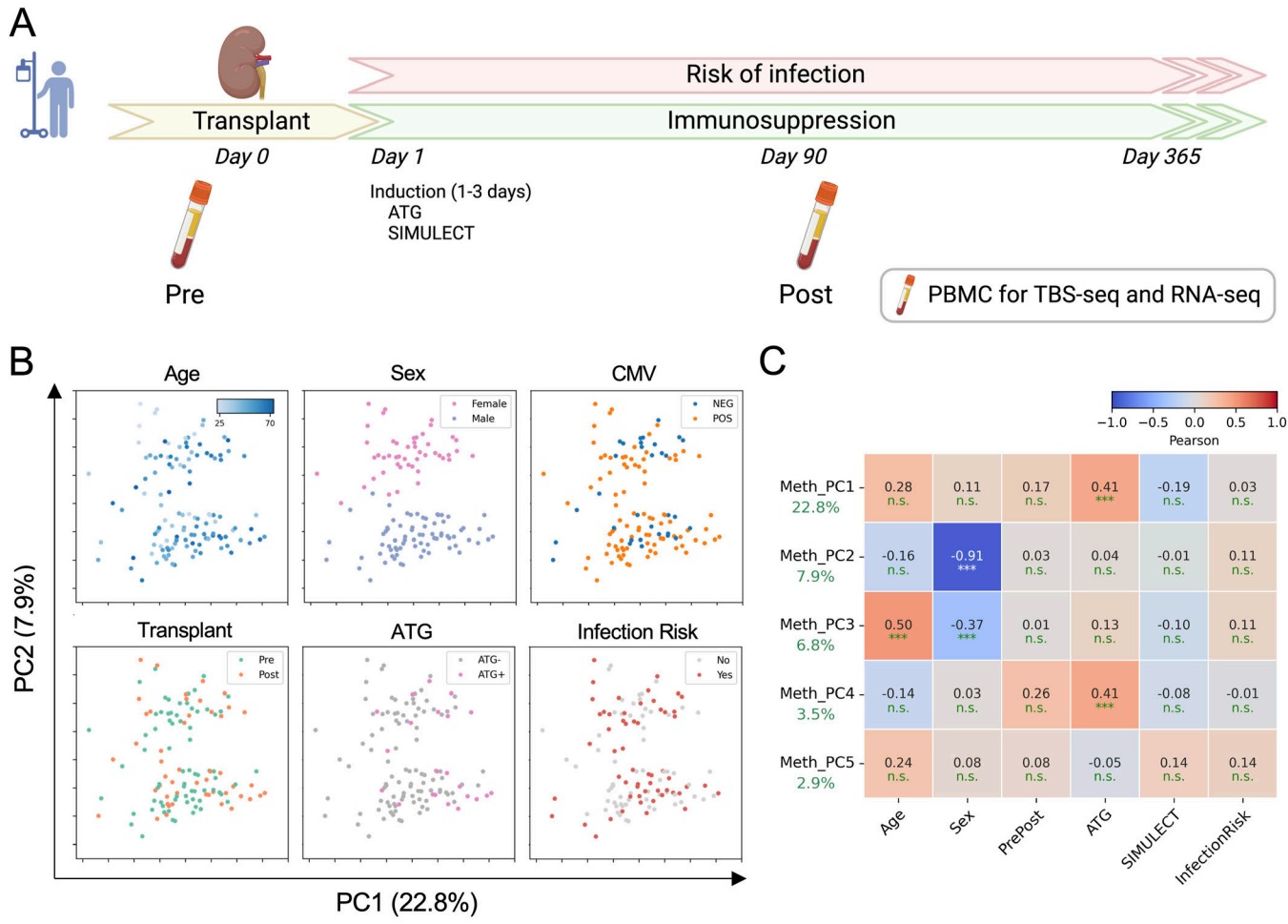

**Figure 1. Study overview.**
**(A)** Schematic workflow of sample collection. Blood was collected before KTx at day 0 (pre-Tx) and then at day 90 (post-Tx). Data on infection and rejection during the first year after KTx were collected. **(B)** DNA methylation PCA color-coded with the demographic and clinical traits. CMV indicates negative or positive recipient serostatus, Transplant indicates pre- or post-Tx, ATG indicates whether patients received ATG at the time of sampling, and infection risk indicates those who did or did not experience infection in the first year after transplant. **(C)** Correlation matrix of the top 5 DNA methylation PCs with each trait.

for age, sex, CMV serostatus, and cell-type composition. 515 CpG *loci* whose methylation levels are significantly positively associated with ATG induction were identified (Fig 3D, Table S1). Gene ontology (GO) analysis shows that these hyper-methylated CpG *loci* are enriched in leukocyte migration, phagocytic cup component, and CD4 receptor binding (Fig 3E). Because hyper-methylation is typically associated with the silencing of the proximal genes, the GO result suggests an overall immunosuppression of CD4-related lymphocytes. An example of a hyper-methylated site near ZBTB7B, a transcription factor (TF) critical for CD4 T-cell commitment, is shown in Fig S3 (Sun et al, 2005). This result reveals epigenetic changes caused by ATG induction are consistent with CD4⁺ T-cell depletion.

However, we found no CpG *loci* significantly associated with categorical determination of development of infection. We then measured whether the predicted epi-infection score has clinical implications by performing time-to-infection analysis using Cox proportional hazard regression. Fig 4A shows that the epi-infection score is a statistically significant covariate predicting the time to infection (HR 3.21, 95% CI 1.06–9.72, *P* < 0.05*). The epi-ATG score,

however, can predict ATG use but is not associated with the time to infection (Fig 4B), which is consistent with the time-to-infection analysis with induction therapies (ATG/SIMULECT, Fig S4).

### Transplant and ATG induction accelerate epigenetic aging

Age-related DNA methylation changes have been shown to affect renal histology and post-Tx allograft fibrosis (Heylen et al, 2019). DNA methylation age is closely associated with Infection Risk (Schaenman et al, 2020). Given the known association between DNA methylation age and infection in KTx recipients, we sought to identify factors that accelerate epigenetic aging.

We used moderation analysis to determine whether certain factors impact the EpiAge derived from the MMLR model in Fig 3. Fig S5 shows that Transplant (Fig S5A), ATG (Fig S5B), Infection Risk (Fig S5C), and CMV serostatus (Fig S5D) all accelerate epigenetic aging, that is, increased the slope in the age regression models, but only Transplant (*P* < 0.05*) and ATG (*P* < 0.05*) were statistically significant moderators.

**Table 1. Patient characteristics.**

| | All subjects | TBS-seq | | RNA-seq |
| --- | --- | --- | --- | --- |
| | | Pre | Post | Pre/Post |
| n (%) | 90 | 72 (80) | 50 (56) | 37/37 (41) |
| Age median [IQR] | 52 (41, 60) | 52 (40, 59) | 50 (41, 59) | 54 (42, 63) |
| Sex, n (%) | | | | |
| Male | 55 (61) | 43 (60) | 30 (60) | 24 (65) |
| Female | 35 (39) | 29 (40) | 20 (40) | 13 (35) |
| CMV, n (%) | | | | |
| Positive | 65 (72) | 49 (68) | 40 (80) | 23 (62) |
| Negative | 25 (28) | 23 (32) | 10 (20) | 14 (38) |
| First transplant, n (%) | 78 (87) | 62 (86) | 43 (86) | 34 (92) |
| Donor type, n (%) | | | | |
| Deceased | 77 (86) | 60 (83) | 39 (78) | 37 (100) |
| Live | 13 (14) | 12 (17) | 11 (22) | 0 (0) |
| Infection, n (%) | 45 (50) | 34 (47) | 24 (48) | 23 (62) |
| Induction, n (%) | | | | |
| ATG | 60 (67) | 44 (61) | 23 (46) | 30 (82) |
| SIMULECT | 30 (33) | 28 (39) | 27 (54) | 7 (19) |
| Maintenance, n (%) | | | | |
| TAC | 86 (96) | 69 (96) | 48 (96) | 34 (92) |
| BELA | 4 (4) | 3 (4) | 2 (4) | 3 (8) |
| Acute rejection, n (%) | 11 (12) | 9 (13) | 7 (14) | 3 (8) |
| Death, n (%) | 0 (0) | 0 (0) | 0 (0) | 0 (0) |

## Gene expression is impacted by ATG and complements DNA methylation alterations

To explore the transcriptional changes associated with KTx, we profiled 37 pairs of pre- and post-Tx RNA-seq (Table 1). Among the top 5 PCs, PC1 and PC3 are negatively correlated with Transplant (Pre/Post) and ATG (Fig S6A and B). This might result from the fact that among the 37 post-Tx samples, 34 have been treated with ATG. Nonetheless, the PCA suggests that KTx and ATG induction alter gene expression in a synergistic manner.

We identified 63 genes up-regulated and 84 genes down-regulated by ATG induction with the criteria of fold change greater than two and adjusted $P < 0.05$ (Fig 5A and Tables S2 and S3). GO analysis shows that the up-regulated genes are enriched in regulation of anti-inflammation cytokine interleukin-10 and immunological synapse formation (Fig 5B). This result is consistent with the effects of ATG as an immunosuppressant.

The down-regulated genes are enriched for vasculature development and cell migration, and the latter is in line with the hyper-methylated CpG GO that leukocyte mobility is attenuated (Figs 3E and 5B).

We further built a gene expression MMLR model, which includes covariates such as age, sex, and cell types, and successfully discriminate ATG induction with an AUC of 0.95 (Fig 5C). In contrast to the DNA methylation MMLR model, the transcriptomic profiles

cannot predict the infection risk (AUC = 0.55) (Fig 5D). This result further strengthens the conclusion that DNA methylation is more significantly associated with infection risk than gene expression.

Next, we sought to characterize genes whose expression is correlated with ATG-associated proximal CpG sites. DNA methylation pattern is correlated with gene expression. The 515 CpG *loci* associated with ATG were mapped to 275 genes (Fig 3D), and those whose expression is correlated with expression (Pearson's $|R| > 0.3$, $P < 0.05$) were characterized. Genes related to immune response such as retinoid acid, NF-κB, cytokine production, and T-cell proliferation were down-related by hyper-methylation (Table 2 and Fig S7A), and virus defense–related genes were up-regulated (Table 2 and Fig S7B). Differential gene expression analysis was performed between pre- and post-Tx (Fig S8A–C and Tables S4 and S5), and the results suggest that ATG has a more profound impact on gene expression than KTx.

## Discussion

Here, we present a KTx cohort study covering 90 individuals, and profiled their PBMC DNA methylation and gene expression. We found that induction therapy with ATG has a profound impact on the methylome (Fig 1B). Comparing with a nondepleting immunosuppressant SIMULECT, we confirmed that naïve T cells and non-naïve CD4⁺, but not non-naïve

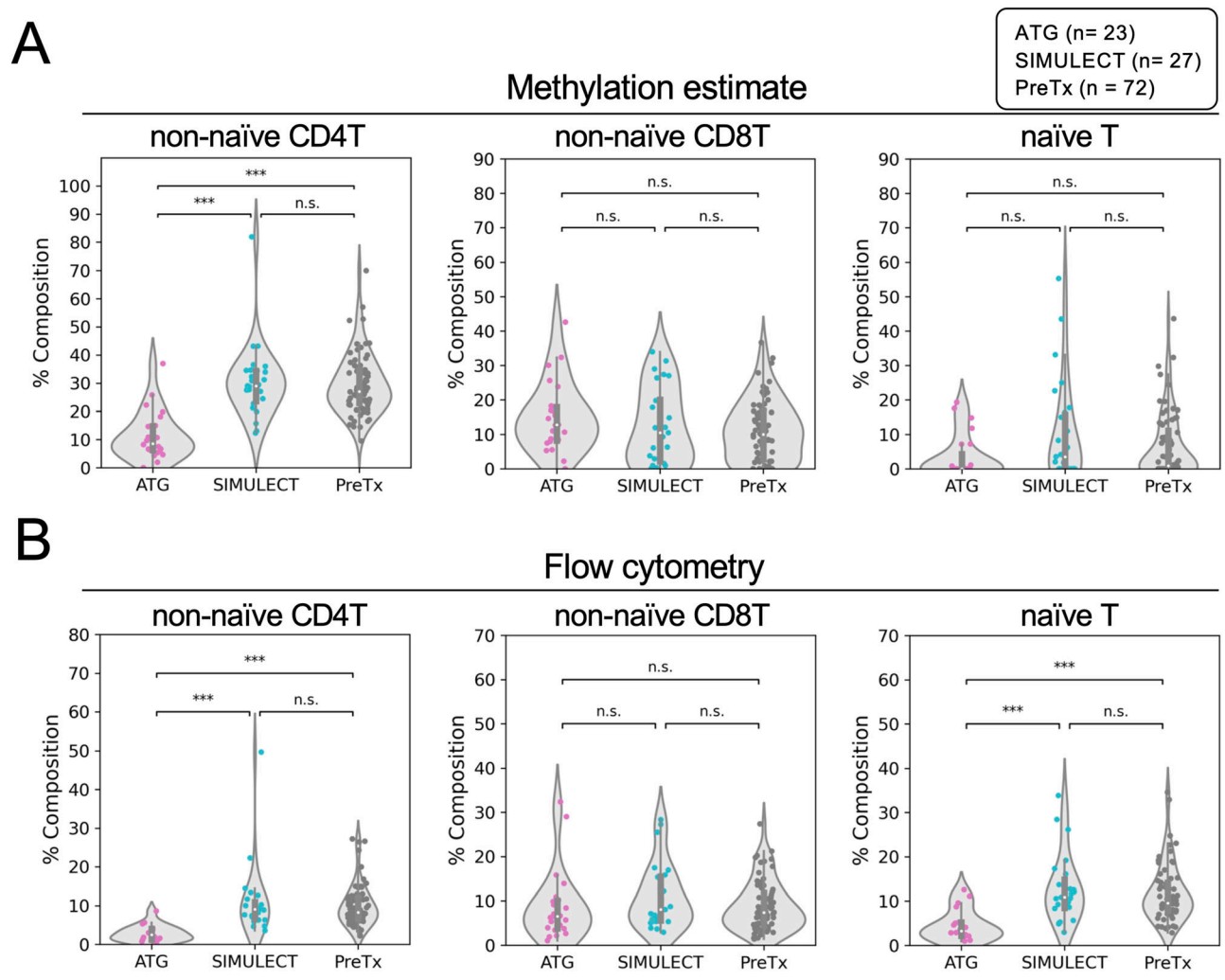

**Figure 2. ATG induction reduces CD4 T-cell population.**
**(A)** Cell composition estimated by DNA methylation. **(B)** Cell composition measured by flow cytometry.

CD8$^+$ T cells, are depleted by ATG, and fail to reconstitute after 3 mo post-Tx (Fig 2). Because there are very few studies in KTx that have measured the effective window of the impact of induction therapies (Gurkan et al, 2010), our study provides a reference for clinicians considering the use of ATG. Future studies can evaluate DNA methylation at 12 mo and beyond to determine whether full immune reconstitution occurs, or whether these depletion-induced changes persist long term after ATG use.

When we corrected for several covariates, we found that (1) ATG induction and (2) infection risk were predictable from the recipients' DNA methylome using an MMLR model. Further statistical analysis identified 515 ATG-associated hyper-methylated CpG *loci* located near genes functioning in leukocyte migration, phagocytosis, and CD4 receptor binding (Fig 3D and E). This observation suggests that the impact of ATG extends beyond simple depletion, leading to functional changes in immune cell function that may explain the long-term impact of ATG.

Although infection was predictable from the recipients' DNA methylome, we were not able to identify individual statistically

significant *loci* associated with infection risk. Nonetheless, the predicted epi-infection score could project time to infection (Fig 4A), whereas the epi-ATG score could not (Fig 4B). This suggests that the methylome harbors widespread but weak signals that are predictive of infection risk. Patients with an epi-infection score suggestive of infection risk could have their medical care adapted in several ways to try to prevent infection after transplantation. Surveillance testing for opportunistic infections such as CMV and BK could be carried out. PCR testing could be extended beyond the conventional period of testing for patients identified to be at high risk. Similarly, antibiotic prophylaxis could be extended for patients identified to be at increased risk. Finally, patients with methylation signals indicative of infection before immunosuppression start could receive individualized immunosuppression such as lower doses of mycophenolate mofetil maintenance immunosuppression.

We have characterized 4 CpG *loci* in DNMT3A hyper-methylated after ATG treatment (Fig 3D and Table S1). This could partially explain why the DNA methylome changes after ATG treatment, but there also could be other indirect mechanisms regulating DNA

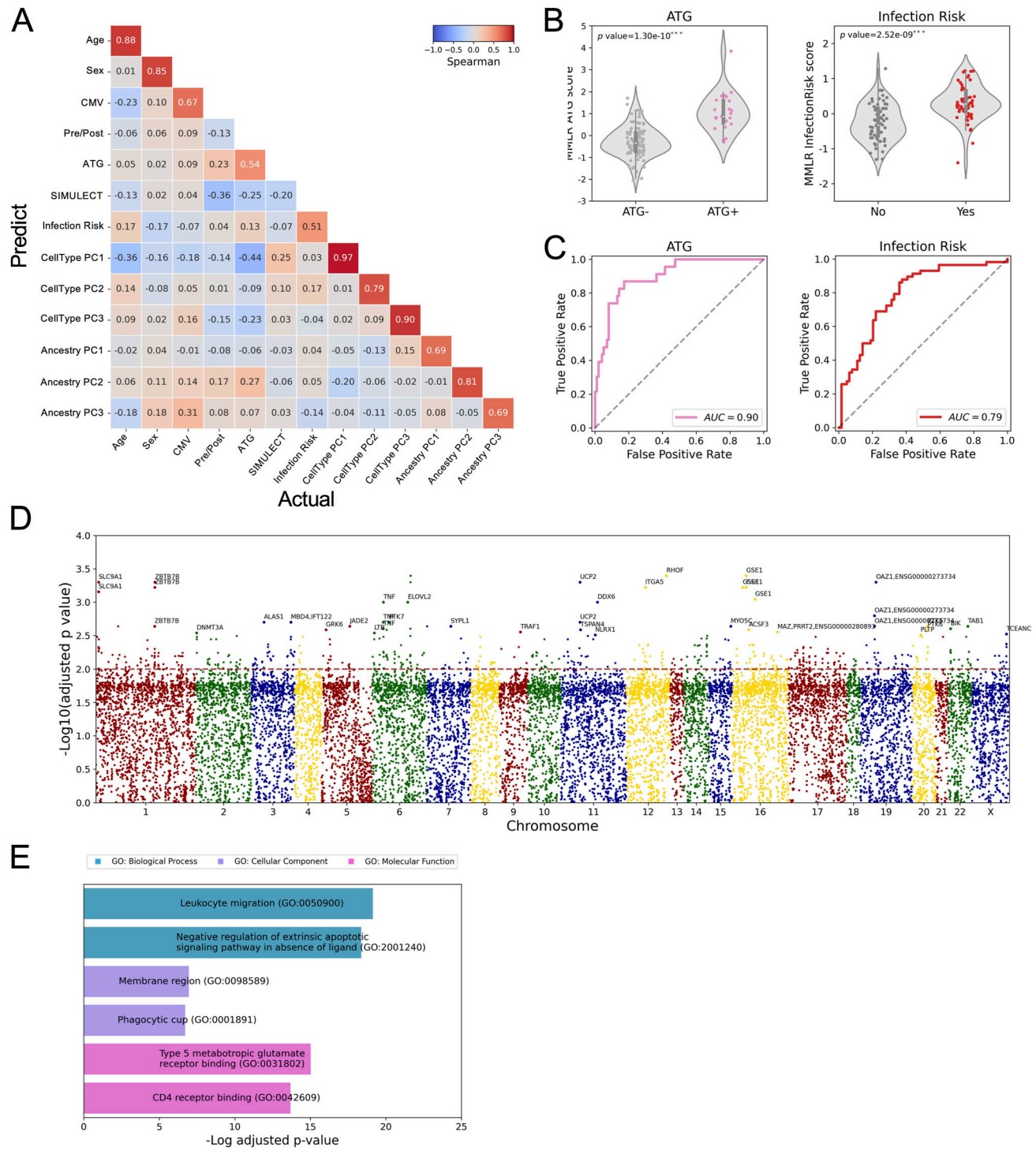

**Figure 3. Multivariate multiple linear regression model.**
**(A)** Spearman correlation matrix of actual-prediction traits. **(B)** Distribution of predicted values of ATG (left) and Infection Risk (right). **(C)** ROC curves of ATG (left) and Infection Risk (right). **(D)** Manhattan plot shows the 515 hyper-methylated CpG sites with ATG treatment. **(E)** Gene ontology of genes covered by the 515 hyper-methylated CpG sites with ATG treatment.

methylation. CD4$^+$ T cells orchestrate immunity and help activate other immune cells, which could include epigenetic reprogramming (Fanucchi et al, 2021). When depleted of CD4$^+$ T cells, the body's ability to respond to infection is severely compromised, and autoimmune reactions may rise; that is, a subset of CD4$^+$ T cells are self-reactive and differentiate into Tregs that help suppress other

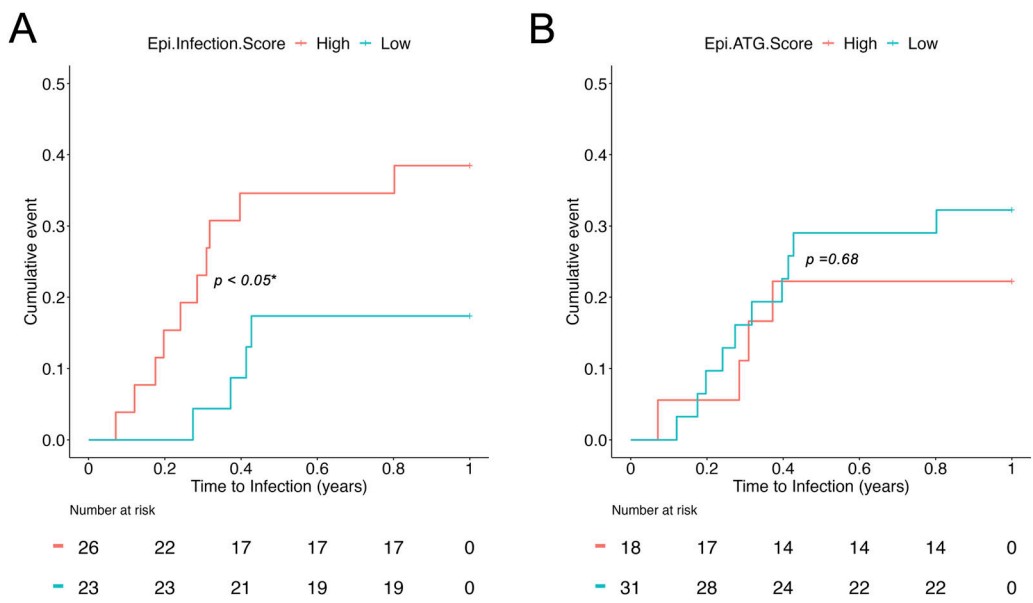

**Figure 4.** **Time-to-infection survival analyses with DNA methylation–estimated covariates.**
**(A)** Epi-infection score from the MMLR model is statistically associated with time to infection. **(B)** Epi-ATG score is not associated with time to infection.

immune cells from attacking the body's own tissues (Marrack & Kappler, 2004; McKinney et al, 2015; Saggau et al, 2024), and evidence has been proposed linking DNA methylation and autoimmune diseases (Ballestar et al, 2020). We observed that immunity-related genes such as FYN, TNF, and IL-6R are hyper-methylated, suggesting dysregulated immunity after ATG treatment. We hypothesize that the cumulative effect of these changes leads to the DNA methylome perturbations we observe.

CKD is an age-related disease that accelerates aging in multiorgan disease networks (Benzing & Schumacher, 2023; Tian et al, 2023). A recent cross-cohort study showed that kidney failure patients have higher epigenetic age than the population-based control, and in a 1-yr window, patients who received KTx had reduced epigenetic age acceleration than the dialysis group (Neytchev et al, 2024). We measured epigenetic aging within our cohort. Our data show that KTx accelerates epigenetic aging, so does the ATG induction (Fig S5). This analysis highlighted the impact of KTx surgery and post-Tx immunosuppression on epigenetic age. Whether the age acceleration will slow down or reverse with longer time of follow-up needs to be investigated.

The transcriptome is altered by ATG induction (Fig 5). We found the ATG-associated hyper-methylated genes are enriched in immune response pathways such as retinoic acid, NF-κB, cytokine production, whereas T-cell proliferation genes are down-regulated. A knockout study in mice suggested that TNFRSF25, a member of TNF receptor superfamily, is associated with T-cell reduction in the thymus (Wang et al, 2001). Our finding suggests that ATG treatment leads to the hyper-methylation of TNFRSF25 and results in CD4[+] T-cell depletion. The down-regulation of CD6's can lead to impaired lymphocyte activation (Gimferrer et al, 2004; Zimmerman et al, 2006). EDAR, which encodes a receptor for ectodysplasin A that can activate NF-κB, was down-regulated through hyper-methylation (Döffinger et al, 2001). These results all support the compromised immunity caused by ATG-induced CD4[+] T-cell depletion.

Finally, the transcriptomic MMLR model can classify ATG induction but not the infection risk (Fig 5C). This difference from the DNA methylation MMLR model further emphasizes the predictive power of DNA methylation.

These results provide an epigenetic characterization of ATG's lymphodepletion capability and potential long-term impact on T-cell function. The ability to use noninvasive testing strategies to analyze the impact of induction therapy and identify patients at increased risk of infection can be leveraged to create tools for individualization of immunosuppression to prevent outcomes of infection and rejection after KTx.

### Limitations of the study

Our study has a few limitations. First, this is a single-center cohort study. The variance of treatment effects could be minimal with the compensation of significance. The second limitation is the lack of full RNA-seq assessment of the cohort that underwent TBS-seq measurement. However, strengths of the findings are the large number of patients evaluated, representing the largest DNA methylation cohort of KTx recipients. Another limitation is the lack of assessment beyond 3 mo post-Tx. Future studies will extend to 6 and 12 mo and beyond to determine the persistence of ATG-induced changes.

# Materials and Methods

### Human subjects

The study procedures, informed consent, and data collection documents were reviewed and approved by the Institutional

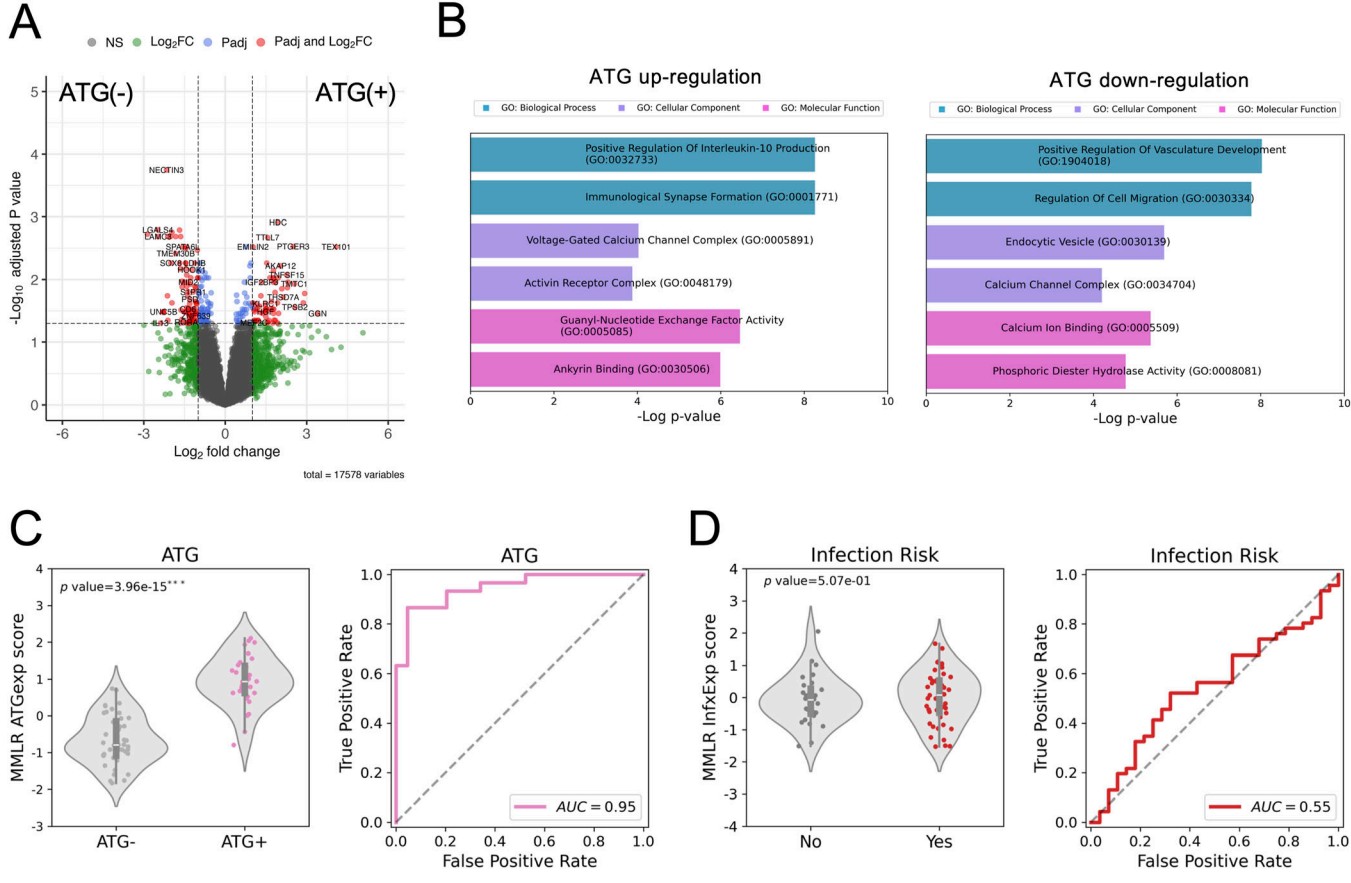

**Figure 5. ATG induction alters gene expression.**
**(A)** Differential gene expression analysis of ATG induction. **(B)** GO analyses of up-regulated (left) and down-regulated (right) genes by ATG induction. **(C, D)** Transcriptomic MMLR model predicts ATG induction (C) but not Infection Risk (D).

**Table 2. Top enriched pathways of gene expression through ATG-induced DNA methylation alterations.**

| Pathway | P | Genes | Direction |
|---|---|---|---|
| Response to Retinoic Acid (GO:0032526) | $1.44 \times 10^{-4}$ | LTK; RARA; PTK6 | Down-regulation |
| Regulation of I-kappaB Kinase/NF-kappaB Signaling (GO:0043122) | $6.23 \times 10^{-4}$ | EDAR; TNFRSF25; TRAF1; LTB | |
| Positive Regulation of Interleukin-12 Production (GO:0032735) | $1.43 \times 10^{-3}$ | IL-23A; LTB | |
| Positive Regulation of Cytokine Production (GO:0001819) | $2.04 \times 10^{-3}$ | CD6; IL-23A; RARA; LTB | |
| Positive Regulation of T Cell Proliferation (GO:0042102) | $5.46 \times 10^{-3}$ | CD6; IL-23A | |
| Regulation of Defense Response To Virus by Host (GO:0050691) | $3.93 \times 10^{-4}$ | APOBEC3G; IFNLR1 | Up-regulation |

Review Board of UCLA (IRB#11-001387). Informed consent was obtained from all participants. A chart review was performed to acquire demographic and clinical data. Participants provided blood samples on the day of transplantation (pre-Tx) and 3 mo after transplantation (post-Tx). Protocol for immunosuppression taper and prophylaxis against infection was described previously (Pickering et al, 2022). This cohort was treated on azathioprine-free regime, and by 3 mo post-Tx, all patients were receiving 5 mg prednisone by mouth daily. Patients with evidence of pre-Tx

sensitization by single antigen testing received ATG induction; other patients received basiliximab. Biopsy was performed for cause, and acute rejection (AR) was defined as biopsy-proven as defined by the Banff criteria (Nickeleit et al, 2018) and the days from Tx to AR are recorded in Table S6. Patient infection includes bacterial (e.g., *Enterococcus faecalis*, *Klebsiella pneumoniae*, *E.coli*, *Clostridium difficile,* and *Staphylococcus epidermidis*), viral (e.g., respiratory syncytial virus, BK virus, cytomegalovirus, herpes simplex virus, rhinovirus, varicella-zoster virus, and COVID-19), and

fungal (e.g., *Candida albicans* and *Aspergillus niger*) infections. Incidence of infection was determined by chart review of the electronic medical record, with infection defined based on Infectious Diseases Society of America criteria as previously described (Schaenman et al, 2021).

## Blood samples

8 ml of blood was drawn into an ACD tube. After Ficoll density gradient centrifugation, PBMCs were separated, isolated, and cryopreserved in FCS/DMSO.

## TBS-seq

### Probe design
The probe panel design is based on the following criteria to include CpG *loci* that (1) cover sites used in DNA methylation clock age estimators (Hannum et al, 2013; Horvath 2013), (2) cover cell type–specific sites, and (3) are located in the promoter regions (–1,000 to +250 bp from TSS) of viral response genes (Morselli et al, 2021). 6,803 biotinylated probes covering the selected CpG *loci* were synthesized by IDT (Integrated DNA Technologies). Probe coordinates are listed in Table S7.

### Library preparation
Genomic DNA was extracted from PBMCs using the phenol–chloroform method (Guha et al, 2018). 500 ng genomic DNA was sheared and subject to end-repair, A-tailing, and ligated with methylated adapters. Purified libraries were hybridized to biotinylated probes and subjected to bisulfite conversion (Cat# D5030; Zymo). Captured DNA was PCR-amplified with KAPA HiFi HotStart Uracil⁺ (Cat# KK2801) into a final TBS-seq library. Library quality was evaluated using TapeStation with the high-sensitivity D1000 tape (Cat# 5067-5584; Agilent). A comprehensive TBS-seq protocol is described in Morselli et al (2021).

### TBS-seq data processing
Cutadapt (Martin, 2011) was used for adapter trimming, and only reads with a minimum of 30 bp were kept for downstream analysis. Reads were aligned to the indexed GRCh38 reference genome using the *BSBolt align* function, and the duplicated reads were marked with the *samtools markdup* function before calling methylation using the *BSBolt callmethylation* function (Farrell et al, 2021). CGmaps from all samples were aggregated into one methylation count matrix using the *BSBolt aggregatematrix* function with parameters *-min-coverage 20 -min-sample 1.0*.

## Cell-type deconvolution

A reference-based cell-type deconvolution approach was used to estimate cell-type composition from DNA methylation profiles (Morselli et al, 2022). To estimate cell-type composition of PBMCs, WGBS datasets from six cell types were used: B cell, CD4 T cell, CD8 T cell, NK cell, naïve T cell, monocyte (from GSE186458 [Loyfer et al, 2023]), and neutrophil band cells (from the BLUEPRINT database [Martens & Stunnenberg, 2013]) (Table S8). Cell type–specific

differentially methylated regions were identified by one-vs-all comparisons using metilene (Jühling et al, 2016) with the criteria to find differentially methylated regions that are (1) at least 500 bp, (2) with the delta methylation level < –30%, and (3) with a false discovery rate < 0.05. Cell type–specific CpG sites were extracted from each TBS-seq sample with the bedtools intersect function and used as input files for deconvolution. Nonnegative least squares regression was used to estimate coefficients.

## Methylation modeling

### MMLR
For each individual $i$, the methylation status of a targeted locus $j$ is denoted as $M_{ij}$. Suppose every $M_{ij}$ is described by $k$ methylation-associated traits, $T_{ik}$, that are weighted by a coefficient $C_{kj}$, the methylation model is formulated as Equation (1):

$$M_{ij} = T_{ik} \times C_{kj} \begin{cases} i \in number\ of\ individuals \\ j \in number\ of\ CpG\ loci \\ k \in number\ of\ traits \end{cases}. \qquad 1$$

This model represents a system of equations in which $T_{ik}$ and $M_{ij}$ are known variables. Our goal is to estimate $C_{kj}$, which represents characteristics of sites, and can be achieved by solving Equation (1) as

$$C_{kj} = T_{ik}^{\dagger} \times M_{ij}. \qquad 2$$

Here, $T_{ik}^{\dagger}$ is derived through the Moore–Penrose pseudoinverse of $T_{ik}$.

### Leave-one-out cross-validation
To avoid overfitting, for each biological sample, a separate MMLR model was trained with the rest samples to derive $C$, and the trait prediction is made as

$$T_{ik} = M_{ij} \times C_{kj}^{\dagger}. \qquad 3$$

Here, $C_{kj}^{\dagger}$ is the Moore–Penrose pseudoinverse of $C_{kj}$.

### Epi scores
For each trait $k$, the individual $i$ will have a predictive value $T_{ik}$ which we term an epi score. For instance, if $k = \begin{cases} 0\ (No\ infection) \\ 1\ (Infection) \end{cases}$, the predicted value $T_{ik}$ is the epi-infection score of this patient. We note that the prediction is independent of the observation; for example, if the patients are not included in this cohort, once their DNA methylation profile is established, their epi scores could still be acquired by the MMLR model trained with this cohort.

### Identification of ATG-associated CpG sites
To identify statistically significant associations between trait and per-site methylation, we estimated the significance of the coefficients using the following equation:

$$y_m = \beta_0 + \beta_1 x_{m1} + \beta_2 x_{m2} + \dots + \beta_n x_{mn} + \varepsilon. \qquad 4$$

Here, $y_m$ is the methylation level at locus $m$, $x_n$ are the explanatory variables including age, sex, cytomegalovirus (CMV) serostatus, ATG induction (Yes/No), Infection Risk (Yes/No), Transplant (Pre/Post), cell-type PCs, and ancestry PCs. $\beta_0$ is the y-intercept, $\beta_n$ is the coefficient for each explanatory variable, and $\varepsilon$ is the error. For each CpG site $y_m$, $P$-values from the model per explanatory variable $x_n$ were derived and adjusted for multiple hypothesis testing using the Benjamini–Hochberg correction. CpG sites with adjusted $P < 0.05$ for ATG (Yes/No) were defined as ATG-associated sites.

### Cox proportional hazards (Coxph) model

Only patients with CMV seropositive status are included in the analysis. The epi-infection score derived from the MMLR prediction of Infection Risk was treated as a covariate of the Coxph model to estimate the rate of infection over the tracking time of 365 d. The Coxph regression analysis was performed with the R package "survival" and plotted with the R package "ggsurvplot."

### Functional enrichment analysis

Site-level GO enrichment analysis was performed using GREAT (McLean et al, 2010) with CMV-associated sites' coordinates as foreground and all TBS-seq–captured CpG sites as background. Gene-level GO enrichment analysis was conducted with Enrichr (Chen et al, 2013).

### Flow cytometry

PBMCs were thawed and stained with fluorochrome-conjugated monoclonal antibodies, then fixed in FluoroFix buffer (Cat# 422101; BioLegend) using standard procedures as previously described (Schaenman et al, 2022). Antibodies used in the CD4[+] and CD8[+] T-cell panels included CD3, CD4, CD8, CD45RA, CCR7, CD57, CD28, KLRG1, and PD1.

### RNA-seq

#### Library preparation

RNA was extracted from PBMCs. mRNA libraries were constructed using KAPA RNA HyperPrep Kit following the manufacturer's instruction (Cat# KK8540). RNA was sheared and primed with oligos for cDNA synthesis. After adapter ligation and PCR amplification, the final library was quantified, and quality was assessed using a TapeStation.

#### RNA-seq data processing

Reads were aligned to reference genome GRCh38 using STAR using default parameters (Dobin et al, 2013). A gene count table was generated using featurecount (Liao et al, 2013). The gene count matrix was first normalized, and DEGs were identified using the R package DESeq2 (Love et al, 2014). Genes with adjusted $P < 0.05$ and at least twofold difference between CMV serostatus groups were considered differentially expressed.

## Data Availability

TBS-seq and RNA-seq data from this study are deposited to Gene Expression Omnibus under the accession number GSE250536.

### Ethics declarations

#### Ethics approval and consent to participate

The study procedures, informed consent, and data collection documents were reviewed and approved by the Institutional Review Board of the UCLA (IRB#11-001387). All procedures were conducted in accordance with the principles outlined in the Declaration of Helsinki.

#### Consent for publication

Informed consent was obtained from all subjects involved in this study.

## Supplementary Information

## Acknowledgements

We thank the UCLA BSCRC Sequencing Core for sequencing the TBS-seq. F-M Hsu is supported by UCLA QCBio Collaboratory and IDRE postdoctoral fellowships.

### Author Contributions

F-M Hsu: data curation, formal analysis, validation, investigation, methodology, and writing—original draft, review, and editing.
H Pickering: resources, data curation, and writing—review and editing.
L Rubbi: resources.
M Thompson: resources.
EF Reed: resources.
M Pellegrini: supervision, investigation, and writing—review and editing.
JM Schaenman: resources, funding acquisition, investigation, and writing—review and editing.

### Conflict of Interest Statement

The authors declare that they have no conflict of interest.

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
