## [Reviewer comments · Life Science Alliance]

Life Science Alliance

DNA methylation predicts infection risk in kidney transplant recipients

Fei-Man Hsu, Harry Pickering, Liudmilla Rubbi, Michael Thompson, Elaine Reed, Matteo Pellegrini, and Joanna Schaeenman
DOI: <https://doi.org/10.26508/lsa.202403124>

Corresponding author(s): Matteo Pellegrini, University of California, Los Angeles

Review Timeline:

Submission Date:	2024-11-04
Editorial Decision:	2025-01-21
Revision Received:	2025-03-04
Editorial Decision:	2025-04-09
Revision Received:	2025-04-15
Accepted:	2025-04-22

Scientific Editor: Tim Fessenden

Transaction Report:

January 21, 2025

Re: Life Science Alliance manuscript #LSA-2024-03124-T

Dr. Matteo Pellegrini
UCLA
MCD Biology
Department of MCD Biology
Los Angeles, CA 90095

Dear Dr. Pellegrini,

Thank you for submitting your manuscript entitled "Anti-thymocyte globulin induction causes activated CD4 and naïve T cell depletion and DNA methylation changes in kidney transplant recipients" to Life Science Alliance. The manuscript has now been seen by expert reviewers, whose reports are appended below. Unfortunately, after an assessment of the reviewer feedback, our editorial decision is against publication in Life Science Alliance.

Given these Reviewer concerns, we are afraid that we are unable to proceed further with the paper. We are thus returning your manuscript to you at this time.

We are sorry our decision is not more positive, but hope that you find the reviews constructive. Of course, this decision does not imply any lack of interest in your work and we look forward to future submissions from your lab.

Thank you for your interest in Life Science Alliance.

Sincerely,

Reviewer #1 (Comments to the Authors (Required)):

The authors have used a highly specific DNA methylation technology on a cohort of kidney transplant recipients to evaluate the effect of a specific therapy on the methylation changes observed and evaluate if some of these methylation changes could be used as prognostic markers. The number of patients studied and the study design are correct for this purpose. DNA methylation is a complex process, to analyse changes in gene sets with a view to predict post-transplant events is a brave endeavour; and should findings be obvious and not controversial it would be a very interesting addition to our academic knowledge.

Transplant recipients are difficult to evaluate as they are generally treated with a number of drugs and their effects are indeed rather difficult to distil as they are administered at the same time, with doses adjusted for different reasons. The first surprise in this article is the absence of information (and therefore analysis) of whether any patient was treated with steroids or azathioprine; both of which have the potential to alter methylation of a large number of genes (PMID: 35146694; PMID 31498880), particularly steroids. In table 1; this information is missing; and if patients were treated on a steroid-free regime this should be highlighted. In table 1 we can also see, that of the patients studied 12% had at least one episode of acute rejection, which generally is treated with high doses of methyl-prednisolone, the timing of this therapy with respect to the second blood sample will be critical to evaluate. At minimum therapy and timing of the acute rejection episodes should be mentioned. "Steroid presence" in the blood sample should be used as confounder variable in all of the statistical analysis, in principle.

The second issue I have missed in the introduction is an explanation of the mechanism of action whereby ATG - a polyclonal antibody that targets membrane receptors and kills many cells (normally by apoptosis, ADCC and complement activation - CDC) but not all of them in peripheral blood, as the authors have observed with the activated CD8 T cells. How would ATG have the ability of affecting a large number of methyl-transferases in order to exert the changes observed in the methylome?. I have read the article and I am still pondering; how? The authors claim that the visualisation of principal component analysis of in Fig 1 clearly show that ATG+ vs ATG- samples can be differentiated. Unless I am not interpreting the clouds correctly; whereas there is a clear differentiation in samples from males and females; the distribution of pink dots (ATG+) and grey dots (ATG-) in the 2 clusters is rather random; I can't see the separation on the PC1 axis mentioned. Whereas the fact that indeed 515 sites seem to

be hypermethylated when comparing ATG vs non-ATG samples is a much stronger signal of the effect of ATG (Fig 3D); and I remain amazed at the number of genes affected. For this, I would have liked, or I think is reasonable to expect, that the authors tried to focus on the affected genes and maybe elaborate a hypothesis on how ATG could be influencing these methylation changes.

Thirdly, the calculation of the predicted epi-infection score is not really well detailed in the method section; somehow it seems that it maybe calculated with some degree of influence of the samples that had infection; therefore the correlation of a high predictive score (Fig 4A) with shorter time to infection, seems at face value like a circular argument. Could the authors clarify how that score is calculated, and whether the prediction is independent of the observation. In the discussion. It should also be mentioned how would this information could be used in the post-transplant care of patients; how can we avoid the infections, is the use of antibiotics or additional anti-virals warranted with such score?

The 4 genes that seem downregulated and hypermethylated specifically in relation to ATG therapy are interesting and a stronger effort in hypothesizing how this finding could come about would have been appreciated.

In my humble opinion, addressing the possible steroid confounding effect on the data is a fundamental pre-requisite for the paper to be publishable. If a hypothesis to link ATG mechanisms of action to large DNA methylation could be added, that would be very wellcome.

Reviewer #2 (Comments to the Authors (Required)):

Short summary of the main findings

Hsu and colleagues applied targeted bisulfite sequencing on total PBMCs isolated from kidney transplant patients prior and after transplantation to measure DNA methylation changes.

Not surprisingly ATG induction therapy in comparison to non-depletional induction therapy induced massive changes in DNA methylation. Also, samples from patients experiencing episodes of infections showed altered DNA methylation profiles.

The scientific advancement is rather minor.

Comments to main points

1) DNA methylation is correlated with ATG induction

- The strongest differences in PCA analysis reflects sex-specific effects along PC2

- No proper statistical analysis is provided which verifies the assumptions that ATG is altering the DNA methylation status of PBMCs

2) ATG induction depletes activated CD4+ and naïve T cell

- the shown flow cytometry results in Figure 2 correspond to the quantification of total and naïve CD4+ T cells but not activated CD4+ T cells

- no major scientific advancements

3) Predicting ATG induction and infection risk with DNA methylation

- it is not described in the text which type of infections were considered, e.g. mainly CMV and herpes virus infections or whether also bacterial infections were considered?

4) Site-specific DNA methylation impact of ATG induction

- no comments

5) Transplant and ATG induction accelerate epigenetic aging

- no specific p value provided, influence seems of transplantation and ATG induction therapy on epigenetic aging seems to be rather minor

6) ATG impacts the transcriptome more than KTx

- not surprising, rather expected and has been shown before

February 3, 2025

Dear Dr. Sawey, PhD,

The authors of manuscript #LSA-2024-03124-T have requested an appeal. Their comments are below.

Dear Editor,

I am writing to express my gratitude for the feedback provided on our manuscript titled "Anti-thymocyte globulin induction causes activated CD4 and naïve T cell depletion and DNA methylation changes in kidney transplant recipients." We appreciate the time and effort that you and the reviewers have dedicated to evaluating our work. We have carefully considered the comments and suggestions made by the reviewers and address them point by point below and also highlighted the revised sections of the manuscript in yellow. Given the fact that we have addressed all of the comments, we hope that you will consider our manuscript for publication. We feel strongly that the work is innovative and has significant potential impact in the field of transplantation.

Reviewer#1 had positive feedback and suggested several modifications before publication. To address the first of these we clarified that no patients were treated with azathioprine in this cohort and all patients were treated with prednisone 5mg daily 3 months post-transplant. To address the second critique, we present a hypothesis that describes how DNA methylation changes might occur due to anti-thymocyte globulin treatment.

Reviewer#2 expressed concerns regarding the significance of our results. To address this, we performed additional analyses to complement the transcriptome analysis we present initially. As a result, we can now claim that this study is the first comprehensive epigenome/transcriptome evaluation of kidney transplant recipients. Moreover, we describe that we were able to develop a DNA methylation biomarker to predict a patient's infection risk.

Because of these changes we kindly request the opportunity to resubmit our manuscript with the revised title "DNA methylation predicts infection risk in kidney transplant recipients". We are confident that the revisions will significantly strengthen our work and align it more closely with the journal's standards.

Thank you for considering our request. We look forward to the possibility of resubmitting our manuscript for consideration.

Sincerely,

Dr. Matteo Pellegrini

February 4, 2025

MS: LSA-2024-03124-T

Dr. Matteo Pellegrini
Department of MCD Biology
UCLA
Los Angeles, CA 90095

Dear Dr. Pellegrini,

Your manuscript entitled "Anti-thymocyte globulin induction causes activated CD4 and naïve T cell depletion and DNA methylation changes in kidney transplant recipients" has now been reconsidered, and I am pleased to let you know that we have decided to send your revised manuscript for external re-review.

Please use the following link to submit your revised manuscript:

<https://lsa.msubmit.net/cgi-bin/main.plex?el=A4Na5BWn6A1teW4I5B9ftde27963oduabu9ZnTUDiBuQZ>

Yours sincerely,

Eric Sawey, PhD
Executive Editor
Life Science Alliance

Reviewer #1 (Comments to the Authors (Required)):

The authors have used a highly specific DNA methylation technology on a cohort of kidney transplant recipients to evaluate the effect of a specific therapy on the methylation changes observed and evaluate if some of these methylation changes could be used as prognostic markers. The number of patients studied and the study design are correct for this purpose. DNA methylation is a complex process, to analyse changes in gene sets with a view to predict post-transplant events is a brave endeavour; and should findings be obvious and not controversial it would be a very interesting addition to our academic knowledge.

1. Transplant recipients are difficult to evaluate as they are generally treated with a number of drugs and their effects are indeed rather difficult to distil as they are administered at the same time, with doses adjusted for different reasons. The first surprise in this article is the absence of information (and therefore analysis) of whether any patient was treated with steroids or azathioprine; both of which have the potential to alter methylation of a large number of genes (PMID: 35146694; PMID 31498880), particularly steroids. In table 1; this information is missing; and if patients were treated on a steroid-free regime this should be highlighted. In table 1 we can also see, that of the patients studied 12% had at least one episode of acute rejection, which generally is treated with high doses of methyl-prednisolone, the timing of this therapy with respect to the second blood sample will be critical to evaluate. At minimum therapy and timing of the acute rejection episodes should be mentioned. "Steroid presence" in the blood sample should be used as confounder variable in all of the statistical analysis, in principle.

We thank the reviewer for the positive comments about our study design and the challenges to evaluate KTx recipients. Regarding steroid/azathioprine treatment, no patients were treated with azathioprine in this cohort. All KTx patients received prednisone 5mg daily after 3 months post KTx in the UCLA medical center. Since this treatment is universal, the prednisone effect does not impact the differential results we presented. We have described the details in Page 6 Line 7-8, Material and Methods (Page 20 Line 13).

2. The second issue I have missed in the introduction is an explanation of the mechanism of action whereby ATG - a polyclonal antibody that targets membrane receptors and kills many cells (normally by apoptosis, ADCC and complement activation - CDC) but not all of them in peripheral blood, as the authors have observed with the activated CD8 T cells.

We have added the description and references regarding the ATG mechanism of action to (Page 3 Line 20-27):

During ATG production, the cell types present in normal human thymus, e.g. thymocytes, were used to inoculate rabbits for the production of thymoglobulin. T lymphocytes (72%) dominate the cell type composition, while B cells (6%), antigen-presenting cells (APC) and stromal cells (22%) contribute to the remainder (Zand et al. 2005). This antibody-dependent lymphodepletion mechanism targets cells expressing antigens in a dosage-dependent manner (Mohty 2007), and it has been reported to cause an inversion of the CD4⁺/CD8⁺ ratio (Mourad et al. 2012).

3. How would ATG have the ability of affecting a large number of methyl-transferases in order to exert the changes observed in the methylome? I have read the article and I am still pondering; how?

The explanation of ATG activity has been added to Discussion in Page 18 Line 3-16: We have characterized 4 CpG loci in DNMT3A hyper-methylated after ATG treatment (Fig.3D and Supplementary Table S1). This could partially explain why the DNA methylome changes following ATG treatment but there also could be other indirect mechanisms regulating DNA methylation. CD4⁺ T cells orchestrate immunity and help activate other immune cells, which could include epigenetic reprogramming (Fanucchi et al. 2021). When depleted of CD4⁺ T cells the body's ability to respond to infection is severely compromised, and autoimmune reactions may rise, *i.e.* a subset of CD4⁺ T cells are self-reactive, and differentiate into Tregs that help suppress other immune cells from attacking the body's own tissues (Marrack and Kappler 2004; Saggau et al. 2024; McKinney et al. 2015) and evidence has been proposed linking DNA methylation and autoimmune diseases (Ballestar et al. 2020). We observed that immunity-related genes such as FYN, TNF and IL6R are hyper-methylated, suggesting dysregulated immunity after ATG treatment. We hypothesize that the cumulative effect of these changes leads to the DNA methylome perturbations we observe.

4. The authors claim that the visualisation of principal component analysis of in Fig 1 clearly show that ATG+ vs ATG- samples can be differentiated. Unless I am not interpreting the clouds correctly; whereas there is a clear differentiation in samples from males and females; the distribution of pink dots (ATG+) and grey dots (ATG-) in the 2 clusters is rather random; I can't see the separation on the PC1 axis mentioned. We added the methylation top 5 PCs' correlation matrix to Figure 1C which shows ATG is correlated with PC1. We hope this quantitative data helps support our conclusion.

Fig. 1

5. Whereas the fact that indeed 515 sites seem to be hypermethylated when comparing ATG vs non-ATG samples is a much stronger signal of the effect of ATG (Fig 3D); and I remain amazed at the number of genes affected. For this, I would have liked, or I think is reasonable to expect, that the authors tried to focus on the affected genes and maybe elaborate a hypothesis on how ATG could be influencing these methylation changes.

This is the same as comment 3.

We have characterized 4 CpG loci in DNMT3A hyper-methylated after ATG treatment (Fig.3D and Supplementary Table S1). This could partially explain why the DNA methylome changes following ATG treatment but there also could be other indirect mechanisms regulating DNA methylation. CD4+ T cell orchestrate immunity and help activate other immune cells, which could include epigenetic reprogramming (Fanucchi et al. 2021) . When depleted of CD4+ T cells the body's ability to respond to infection is severely compromised, and autoimmune reactions may rise , i.e. a subset of CD4+ T cells are self-reactive, and differentiate into Tregs that help suppress other immune cells from attacking the body's own tissues (Marrack and Kappler 2004; Saggau et al. 2024; McKinney et al. 2015) and evidence has been proposed linking DNA methylation and autoimmune diseases (Ballestar et al. 2020) . We observed that immunity-related genes such a FYN, TNF and IL6R are hyper-methylated, suggesting dysregulated immunity after ATG treatment. We hypothesize that the cumulative effect of these changes leads to the DNA methylome perturbations we observe.

6. Thirdly, the calculation of the predicted epi-infection score is not really well detailed in the method section; somehow it seems that it maybe calculated with some degree of influence of the samples that had infection; therefore the correlation of a high predictive score (Fig 4A) with shorter time to infection, seems at face value like a circular argument. Could the authors clarify how that score is calculated, and whether the prediction is independent of the observation.

We added details regarding the epi score of each trait on Page 23 Line 7-11, For each trait k , the individual i will have a predictive value T_{ik} which we term an epi score. For instance, if $k = \begin{cases} 0 & (No\ infection) \\ 1 & (Infection) \end{cases}$, the predicted value T_{ik} is the epi-infection score of this patient. We note that the prediction is independent of the observation, e.g. if the patients are not included in this cohort, once their DNA methylation profile is established, their epi scores can be measured by the MMLR model trained with this cohort.

7. In the discussion. It should also be mentioned how would this information could be used in the post-transplant care of patients; how can we avoid the infections, is the use of antibiotics or additional anti-virals warranted with such score?

We have added the sentences below to the Discussion from Page 17 Line 23 to Page 18 Line 2:

Patients with an epi-infection score suggestive of infection risk could have their medical care adapted in several ways to try to prevent infection after transplantation. Surveillance testing for opportunistic infections such as CMV and BK could be carried out. PCR testing could be extended beyond the conventional period of testing for patients identified to be at high risk. Similarly, antibiotic prophylaxis could be extended for patients identified to be at increased risk. Finally, patients with methylation signals indicative of infection prior to immunosuppression start could receive individualized immunosuppression such as lower doses of mycophenolate mofetil maintenance immunosuppression.

8. The 4 genes that seem downregulated and hypermethylated specifically in relation to ATG therapy are interesting and a stronger effort in hypothesizing how this finding could come about would have been appreciated.

We have added the hypothesized mechanism by ATG from Page 18 Line 27 to Page 19 Line 8:

The transcriptome is altered by ATG induction (Fig.5). We found the ATG-associated hyper-methylated genes are enriched in immune response pathways such as retinoic acid, NF- κ B, cytokine production, while T cell proliferation genes are down-regulated. A knockout study in mice suggested that TNFRSF25, a member of TNF-receptor superfamily, is associated with T cell reduction in the thymus (Wang et al. 2001). Our finding suggests that ATG treatment leads to the hyper-methylation of TNFRSF25 and results in CD4⁺ T cell depletion. The down-regulation of CD6's can lead to impaired lymphocyte activation (Gimferrer et al. 2004; Zimmerman et al. 2006). EDAR, which encodes a receptor for ectodysplasin A that can activate NF- κ B, was down-regulated through hyper-methylation (Döffinger et al. 2001). These results all support the compromised immunity caused by ATG-induced CD4⁺ T cell depletion.

In my humble opinion, addressing the possible steroid confounding effect on the data is a fundamental pre-requisite for the paper to be publishable. If a hypothesis to link ATG mechanisms of action to large DNA methylation could be added, that would be very wellcome.

Reviewer #2 (Comments to the Authors (Required)):

Short summary of the main findings

Hsu and colleagues applied targeted bisulfite sequencing on total PBMCs isolated from kidney transplant patients prior and after transplantation to measure DNA methylation changes.

Not surprisingly ATG induction therapy in comparison to non-depletional induction therapy induced massive changes in DNA methylation. Also, samples from patients experiencing episodes of infections showed altered DNA methylation profiles.

The scientific advancement is rather minor.

Comments to main points

1. DNA methylation is correlated with ATG induction
 - The strongest differences in PCA analysis reflects sex-specific effects along PC2
This is a feature of DNA methylation. We have added a reference on this in Page 8 Line 2-5:
In addition to age and sex which are known to be associated with DNA methylation (Gatev et al. 2021) , we found that ATG induction is correlated with the PC1 axis, whereas Transplant (Pre/Post) and Infection were not correlated with the top PCs (Fig.1B and 1C).
 - No proper statistical analysis is provided which verifies the assumptions that ATG is altering the DNA methylation status of PBMCs
We added the methylation top 5 PCs' correlation matrix to Fig.1C which shows ATG is correlated with PC1:

Fig. 1

2. ATG induction depletes activated CD4+ and naïve T cell
 - The shown flow cytometry results in Figure 2 correspond to the quantification of total and naïve CD4+ T cells but not activated CD4+ T cells

Fig.2 shows the non-naïve CD4T, non-naïve CD8T and naïve T cell fraction estimated by DNA methylation (Fig.2A) and flow cytometry (Fig.2B). We have corrected the annotation to be non-naïve CD4T, non-naïve CD8T and naïve T. We note that we did not separate naïve T cells into CD4/CD8 fractions. Both approaches show the reduction in non-naïve CD4T and naïve T. Fig.S1B indicates the correlation coefficients are 0.79 for non-naïve CD4T, 0.49 for non-naïve CD8T and 0.83 for naïve T.

Fig. 2

- No major scientific advancements
 The scientific advancement here is to show we can capture the lymphodepletion of ATG by DNA methylation (TBS-seq) and the results are consistent with the conventional flow cytometry.
- 3. Predicting ATG induction and infection risk with DNA methylation
 - it is not described in the text which type of infections were considered, e.g. mainly CMV and herpes virus infections or whether also bacterial infections were considered?
 We have added the type of infections considered to Page 20 Line 17-21, Patient infection includes bacterial (e.g. Enterococcus faecalis, Klebsiella pneumonia, E.coli, Clostridium difficile and Staphylococcus epidermidis), viral (e.g. Respiratory syncytial virus, BK virus, Cytomegalovirus, Herpes simplex virus, rhinovirus, varicella-zoster virus and COVID-19) and fungal (e.g. Candida albicans and Aspergillus niger).
- 4. Site-specific DNA methylation impact of ATG induction
 - no comments
- 5. Transplant and ATG induction accelerate epigenetic aging

- no specific p value provided, influence seems of transplantation and ATG induction therapy on epigenetic aging seems to be rather minor
We moved this result to Supplemental Fig.S5 and the p values were provided.
6. ATG impacts the transcriptome more than KTx
- not surprising, rather expected and has been shown before
We have re-written this paragraph with additional results to focus on the integration of DNA methylome and transcriptome in Page 13-14:

Gene expression is impacted by ATG and complements DNA methylation alterations

To explore the transcriptional changes associated with KTx, we profiled 37 pairs of pre- and post-Tx RNA-seq (Table 1). Among the top 5 PCs, PC1 and PC3 are negatively correlated with Transplant (Pre/Post) and ATG (Supplemental Fig.S6). This might result from the fact that among the 37 post-Tx samples, 34 have been treated with ATG. Nonetheless the PCA analysis suggests that KTx and ATG induction alter gene expression in a synergistic manner.

We identified 63 genes up-regulated and 84 genes down-regulated by ATG induction with the criteria of fold change greater than 2 and adjusted p value < 0.05 (Supplemental Fig.4A and Supplemental Table S2-3). GO analysis shows that the up-regulated genes are enriched in regulation of anti-inflammation cytokine interleukin-10 and immunological synapse formation (Fig.5B). This result is consistent with the effects of ATG as an immunosuppressant.

The down-regulated genes are enriched for vasculature development and cell migration and the latter is in line with the hyper-methylated CpG GO that leukocyte mobility is attenuated (Fig.5B and Fig.3E).

We further built a gene expression MMLR model which includes covariates such as age, sex and cell types and successfully discriminate ATG induction with an AUC 0.95 (Fig.5C). In contrast to the DNA methylation MMLR model, the transcriptomic profiles cannot predict the infection risk (AUC = 0.55) (Fig.5D). This result further strengthens the conclusion that DNA methylation is more significantly associated with infection risk than gene expression.

Next, we sought to characterize -genes whose expression is correlated with ATG associated proximal CpG sites. The 515 CpG loci associated with ATG were mapped to 275 genes (Fig. 3D) and those whose expression is correlated with expression (Pearson's $|R| > 0.3$, $p < 0.05$) were characterized. Genes related to immune response such as retinoid acid, NF- κ B, cytokine production and T cell proliferation were down-related by hyper-methylation, and virus defense related genes were up-regulated (Table 2 and Supplemental Fig.S7).

April 9, 2025

RE: Life Science Alliance Manuscript #LSA-2024-03124-TR-A

Dr. Matteo Pellegrini
University of California, Los Angeles
MCD Biology
Department of MCD Biology
UCLA
los angeles, ca 90095

Dear Dr. Pellegrini,

Thank you for submitting your revised manuscript entitled "DNA methylation predicts infection risk in kidney transplant recipients" to LSA. As you will see, Reviewer 1 commends the improvements made and recommends publication with only minor changes requested. We would be happy to publish your paper in Life Science Alliance pending those minor changes and final revisions necessary to meet our formatting guidelines.

- please be sure that the authorship listing and order is correct.
- please upload your main manuscript text as an editable doc file;
- please upload your main and supplementary figures as single files.
- please add ORCID ID for corresponding (and secondary corresponding) author- you should have received instructions on how to do so.
- please add the X and Bluesky handles of your host institute/organization as well as your own or/and one of the authors in our system.
- please remove highlights from the text and upload a clean manuscript file
- please upload your Tables in editable .doc or excel format. They can be included at the bottom of the main manuscript file or be sent as separate files.
- please remove figures and their legends from the manuscript file and leave them uploaded separately.
- please incorporate any points from the Conclusion section into the Discussion; we only allow a Discussion section.
- please remove "Authors and Affiliations" information from page 25 and leave only authors' contributions.
- please add your main, supplementary figure, and table legends in the main manuscript text after the references section.
- please add callouts for Figures S1A-B; S5A-D; S6A-B; S7A-B; S8A-C and tables S4; S5.

A. FINAL FILES:

- An editable version of the final text (.DOC or .DOCX) is needed for copyediting (no PDFs).
- High-resolution figure, supplementary figure and video files uploaded as individual files: See our detailed guidelines for preparing your production-ready images, <https://www.life-science-alliance.org/authors>
- Summary blurb (enter in submission system): A short text summarizing in a single sentence the study (max. 200 characters

including spaces). This text is used in conjunction with the titles of papers, hence should be informative and complementary to the title. It should describe the context and significance of the findings for a general readership; it should be written in the present tense and refer to the work in the third person. Author names should not be mentioned.

B. MANUSCRIPT ORGANIZATION AND FORMATTING:

Sincerely,

Reviewer #1 (Comments to the Authors (Required)):

The authors have thoroughly attended each of the comments, of both reviewers actually, and modified parts of the manuscript, added further clarification on the data and incorporated important arguments in the discussion. It seems a stronger paper for publication now.

Minor notes to authors:

1 - It might be relevant to state the timings of the rejection episodes post-Tx, though the information about immunosuppression is a bit better.

2 - in the merged version of the pdf, in the legend of Figure 1; please modify, as you have correlated the top 5 PCs with factors and not just 4.

Reviewer 1

Minor notes to authors:

1 - It might be relevant to state the timings of the rejection episodes post-Tx, though the information about immunosuppression is a bit better.

We have stated the timing of rejection and immunosuppression in Page 14 Line 5-9.

2 - in the merged version of the pdf, in the legend of Figure 1; please modify, as you have correlated the top 5 PCs with factors and not just 4.

We have corrected the legend of Figure 1.

April 22, 2025

RE: Life Science Alliance Manuscript #LSA-2024-03124-TRR

Matteo Pellegrini
University of California, Los Angeles
Department of MCD Biology
los angeles, CA 90095

Dear Dr. Pellegrini,

Thank you for submitting your Research Article entitled "DNA methylation predicts infection risk in kidney transplant recipients". It is a pleasure to let you know that your manuscript is now accepted for publication in Life Science Alliance. Congratulations on this interesting work.

DISTRIBUTION OF MATERIALS:

Again, congratulations on a very nice paper. I hope you found the review process to be constructive and are pleased with how the manuscript was handled editorially. We look forward to future exciting submissions from your lab.

Sincerely,
